# Seroprevalence of Neutralizing Antibodies in Healthy Adults, in Mexico, Against Human and Simian Adenovirus Types

**DOI:** 10.3390/v17091184

**Published:** 2025-08-29

**Authors:** Raúl E. López, Margarita Valdés Alemán, Jesús M. Torres-Flores, Yordanis Pérez-Llano, David Alejandro Cabrera Gaytán, Clara Esperanza Santacruz Tinoco, Julio Elias Alvarado Yaah, Yu Mei Anguiano Hernández, Bernardo Martínez Miguel, José Esteban Muñoz Medina, Nancy Sandoval Gutiérrez, Ilse Ramos Lagunes, José Antonio Arroyo Pérez, Ramón A. González

**Affiliations:** 1Centro de Investigación en Dinámica Celular, Instituto de Investigación en Ciencias Básicas y Aplicadas, Universidad Autónoma del Estado de Morelos, Cuernavaca 62209, Mexico; raul.lopez@ibt.unam.mx (R.E.L.); valdes.aleman.margarita@gmail.com (M.V.A.); yordaniz.perezlla@uaem.edu.mx (Y.P.-L.); 2Laboratorio Nacional de Vacunología y Virus Tropicales, Escuela Nacional de Ciencias Biológicas, Instituto Politécnico Nacional, Ciudad de México 11340, Mexico; jtorresf@ipn.mx; 3Coordinación de Calidad de Insumos y Laboratorios Especializados, Instituto Mexicano del Seguro Social, Ciudad de México 07760, Mexico; david.cabrerag@imss.gob.mx (D.A.C.G.); clara.santacruz@imss.gob.mx (C.E.S.T.); julio.alvaradoy@imss.gob.mx (J.E.A.Y.); yu.anguiano@imss.gob.mx (Y.M.A.H.); bernardo.martinezm@imss.gob.mx (B.M.M.); jose.munozm@imss.gob.mx (J.E.M.M.); nancy.sandovalg@imss.gob.mx (N.S.G.); 4Dirección Técnica y de Investigación, Centro Nacional de la Transfusión Sanguínea, Ciudad de México 17360, Mexico; ilse.ramos@salud.gob.mx (I.R.L.); arroyoqfi@gmail.com (J.A.A.P.)

**Keywords:** adenovirus seroprevalence, healthy Mexican adults, adenovirus neutralizing antibodies, COVID-19 vaccines

## Abstract

Replication-defective adenoviruses are widely used as vectors for vaccines, but their efficacy may be compromised by the prevalence of pre-existing neutralizing antibodies from natural infections or prior vaccination with adenovirus-based vaccines. To overcome these limitations, less common human adenovirus (HAdV) types and simian adenoviruses (SAdV) have been explored as alternative vectors to the widely prevalent HAdV-C5. Despite their importance, there is limited information on the epidemiology of adenovirus immunity in many countries and geographical regions, including Mexico. In this study, we analyzed 2488 serum samples from healthy adults across all 32 states of Mexico to assess the prevalence of both total and neutralizing antibodies against various HAdV types from species A-F, and three related SAdVs. Our findings indicate a high prevalence of neutralizing antibodies against HAdV-C5 and HAdV-C6, with significant cross-reactivity observed among related adenoviruses. Notably, HAdV-D26 exhibited a lower prevalence of neutralizing antibodies, suggesting its potential suitability as a vector for vaccine development in populations with high pre-existing immunity to more common HAdV types. These results provide critical insights for optimizing adenovirus-based vaccine strategies in Mexico.

## 1. Introduction

Adenoviruses (AdV) are non-enveloped, icosahedral, double-stranded DNA viruses that are widely distributed and cause respiratory, gastrointestinal, conjunctival, and other diseases [1,2]. Due to their stability, high production titers, broad cell tropism, and ability to induce strong B- and T-cell responses, they have long been considered excellent candidates as viral vectors for gene therapy, anti-cancer treatments, and vaccines against infectious agents [3,4,5]. However, the effectiveness of human adenovirus (HAdV) vectors for transgene delivery [6,7,8,9] and immune response induction [10,11,12,13], particularly of those based on species C human adenovirus type 5 (HAdV-C5), is often compromised by high levels of pre-existing immunity in the population. This has prompted the exploration of alternative HAdV types with reduced seroprevalence or AdV from non-human primates [4]. The availability of over 200 non-human adenovirus types and more than 110 HAdV types offers substantial potential for designing diverse and effective viral vectors.

A variety of HAdV-based vaccine vectors have been developed targeting diverse pathogens, including human immunodeficiency virus (HIV), tuberculosis (TB), Zika virus, malaria, respiratory syncytial virus (RSV), Ebola, and notably many of the vaccines that have been used against SARS-CoV-2 [14,15,16,17]. In Mexico, four of the eight vaccines that were approved against COVID-19 are based on AdVs, namely: Gam-COVID-Vac (Sputnik V from Gamaleya National Center), which uses HAdV-C5 and HAdV-D26; Ad5-nCoV Convidecia (from Cansino Biologics Inc), based on HAdV-C5; Ad26. COV2-S (from Janssen-Cilag), based on HAdV-D26; and AZD1222 Covishield (from Oxford-AstraZeneca), based on a chimpanzee AdV (ChAdOx-1-2) [18].

The seroprevalence against HAdV has been analyzed in various geographical regions and populations, revealing that the prevalence of different virus types varies depending on multiple factors such as the country’s development, and population size and distribution [19]. Studies consistently show that a significant portion of the human population has been exposed to HAdV-C5, leading to high levels of neutralizing antibodies against this virus type [19,20,21,22,23,24,25,26,27,28]. In contrast, the seroprevalence of HAdV-D26, while initially low in North America and Western Europe, is found to be particularly high in some sub-Saharan populations and varies significantly across different global regions [20,24,25,27]. Moreover, there is insufficient evidence to suggest that the level of long-lasting protection induced by any of the human or simian AdV-based vaccines is uniformly effective across all populations and regions [29].

There is a significant gap in research regarding the prevalence and diversity of circulating adenoviruses in Latin America [30]. In Mexico, available data on seroprevalence against AdVs are scarce and sporadic. It remains uncertain whether the existing prior immunity in the Mexican population, in particular against the types that have been used in COVID-19 vaccines, might impact their efficacy, the duration of immunity, or their potential suitability for vaccine boosters or seasonal use. In this study, we determined the seroprevalence of binding and neutralizing antibodies in healthy adults across all 32 states in Mexico against HAdVs types from species A to F, along with three phylogenetically related simian adenoviruses (SAdVs).

## 2. Materials and Methods

### 2.1. Human Serum Samples

Serum samples were obtained from blood donors over 18 years of age who met the criteria for blood donation, which were those established by the Mexican Official Norm (NOM-253-SSA1-2012) and the official procedures for attention to blood donors at the Blood Banks of the Mexican Institute of Social Security (Number. 2430-003-001, dated 24 November 2020), for which the minimum requirements were (i) official personal identification; (ii) age between 18 and 65; (iii) body weight above 50 kg; (iv) at least 4 h but not more than 8 h of fasting; (v) not having cough, flu, diarrhea, or dental infection in the last 14 days; (vi) no medication in the last 5 days; (vii) not having had endodontics, tattoos, piercings, or acupuncture in the last 12 months; (viii) not having undergone any type of surgery in the last 6 months; (ix) not having been vaccinated in the last 30 days; and (x) not having consumed alcoholic beverages in the last 72 h. Remaining serum samples in primary containers such as cryotubes or tubes in good condition were kept in the cold chain at temperatures between −20 °C and −80 °C.

Two sets of serum samples were analyzed in this study. The first set consisted of pre-COVID-19-pandemic serum samples, obtained from the National Blood Transfusion Center (CNTS) in Mexico City. These samples were collected from healthy adult donors (n = 988) between September 2018 and February 2019, before the first reported case of SARS-CoV-2 in Mexico. This set provides a baseline of adenovirus seroprevalence prior to any potential cross-reactivity with COVID-19-related exposures or vaccines.

The second set comprised pre-vaccination serum samples, collected from healthy adult donors (*n* = 1500) across all 32 states of Mexico between February and August 2020, prior to the availability of COVID-19 vaccines. These samples were obtained from the Mexican Institute of Social Security (IMSS), and they reflect the immunological landscape of the population immediately before COVID-19 vaccination campaigns began.

### 2.2. Cells and Viruses

The adenoviruses used in this study were obtained from the American Type Culture Collection (ATCC), including HAdV-A12 (VR-863), HAdV-B14 (VR-15), HAdV-C6 (VR-6), HAdV-D36 (VR-1610), HAdV-D26 (VR-224), HAdV-E4 (VR-1572), HAdV-F41 (VR-930), SAdV-21 (VR-20), SAdV-25 (VR-594) and SAdV-31 (VR-204), except for HAdV-C5 (H5pg2250, derived from H5pg4100 [31]). The virus types were selected in order to include at least one type from each HAdV species and related SAdV species, and that were immediately available from the ATCC at the time the study was initiated. Consequently, various relevant HAdVs, such as the species G type 52, species B1 types 3 and 7, and species B2 type 11, were not included in the study. All AdVs were propagated in HEK-293 or HeLa monolayers cells (obtained from ATCC. Manassas, VA, USA; catalog number CRL-1573 and CCL-2, respectively). Viral titers were determined by standard plaque assays to quantify infectivity levels. The cells were cultured in Dulbecco’s Modified Eagle Medium (DMEM Gibco, Thermo Fisher Scientific, Waltham, MA, USA), supplemented with 10% fetal bovine serum (Biowest, Nuaillé, France), 100 IU/mL penicillin, and 100 µg/mL streptomycin. The choice of cell line (HEK-293 or HeLa) was dependent on optimal viral propagation conditions for each specific adenovirus type to ensure consistent viral growth.

### 2.3. Anti-AdV IgG Enzyme-Linked Immunosorbent Assay

The presence of antibodies against AdVs was determined with an in-house enzyme-linked immunosorbent assay (ELISA) using AdV-infected cell lysates as the antigen. Optimal antigen concentration was established by performing a titration experiment using twelve serial 1:3 dilutions of a lysate derived from HAdV-C5-infected cells, starting at a 1:25 dilution from cells infected with a viral titer of 1 × 10^9^ PFU/mL. This initial testing used an anti-adenovirus antibody (RayBiotech Inc., DS-MB-00026. Peachtree Corners, GA, USA) at a 1:2500 dilution to determine the best signal-to-background ratio.

After standardization, high-affinity 96-well ELISA plates (Corning Inc., E1A/RIA. Corning, NY, USA) were coated with adenoviral lysates at a 1:4000 dilution from stocks with a viral titer of 1 × 10^8^ PFU/mL, using 50 µL per well. Plates were incubated overnight at 4 °C to ensure optimal antigen binding. The plates were washed three times with 150 µL of 0.1% PBS-Tween 20 (PBS-T) to remove unbound antigen, followed by blocking with 100 µL of 5% milk (Svelty, Nestlé, Vevey, Switzerland) in PBS-T for 2.5 h at room temperature. Plates were washed again as described above.

Serum samples were heat-inactivated at 56 °C for 1 h. Initial tests of serum concentrations were performed using eight serial 1:2 dilutions starting at a 1:25 dilution. Based on these results, the optimal dilution for testing was determined to be 1:100, providing a good signal-to-noise ratio. Sera were subsequently diluted 1:100 in 1% milk in PBS-T, and 50 µL of this dilution was added to each well, followed by a 2 h incubation at room temperature. The plates were then washed three times with PBS-T, and 50 µL of goat anti-human IgG conjugated to horseradish peroxidase (HRP) (Bio-Rad Laboratories Inc., 5172-2504. Hercules, CA, USA) at a 1:7000 dilution was added to each well and plates were incubated for 1 h at room temperature. After washing three times with PBS-T, 100 µL of OPD substrate buffer (containing o-phenylenediamine at 0.4 g/L, citric acid at 4.7 g/L, Na_2_HPO_4_ at 7.3 g/L) was added, with 3% hydrogen peroxide (6 µL/mL of OPD buffer) added immediately prior to use. Plates were incubated in the dark for 10 min, and the reaction was stopped by adding 50 µL of 3N HCl to each well. Absorbances were read at 492 nm on a Thermo Scientific Multi Skan Sky high plate reader. All assays were conducted in duplicate to ensure consistency, and a set of negative control wells (containing non-infected cell lysate) and positive control sera were included in each plate.

### 2.4. Adenovirus Neutralization Assay

Neutralizing antibodies against HAdV and SAdV were assessed using a standard in vitro neutralization assay. HeLa cells were seeded at a density of 4 × 10^4^ cells per well in 96-well plates and cultured overnight. Human sera were heat-treated at 56 °C for 1 h and then diluted in DMEM. Initially, serum neutralization capacity was evaluated using eight serial 1:2 dilutions starting at 1:50, reaching final dilutions up to 1:12,800. Based on these initial tests, the optimal dilution for neutralization was determined to be 1:200, which resulted in 100% inhibition of infection as determined by immunofluorescence using an anti-adenovirus antibody (RayBiotech Inc., DS-MB-00026. Peachtree Corners, GA, USA).

The sera from the study participants were subsequently pooled into 40 groups, based on their ELISA absorbance values for each virus, ranging from the highest to the lowest binding antibody levels. Virus-serum mixtures were prepared by incubating each virus with the pooled sera in DMEM at a multiplicity of infection (MOI) of 5 PFU per cell, which corresponds to >99% tissue culture infectious dose, for 1 h at 37 °C. The virus-sera mixtures were then added to the 96-well plates containing HeLa cells and incubated at 37 °C for 1 h to allow viral adsorption, after which the inoculum was replaced with DMEM containing 10% fetal bovine serum. Twenty-four hours post-infection (hpi), the cells were fixed and analyzed by immunofluorescence microscopy. Cells were washed twice with phosphate-buffered saline (PBS) and then fixed with 3.7% formaldehyde for 20 min at room temperature. Following fixation, the cells were washed again twice with PBS and permeabilized with 50 µL of 0.5% Triton X-100 for 5 min at room temperature. After permeabilization, the plates were washed again with PBS and incubated overnight at 4 °C with the primary anti-adenovirus antibody (RayBiotech Inc., DS-MB-00026. Peachtree Corners, GA, USA) at a 1:1500 dilution. Subsequently, the plates were washed and incubated for 2 h at room temperature with a secondary antibody conjugated to Alexa Fluor 488. To counterstain cell nuclei, DAPI was added at a 1:20,000 dilution. Micrographs of 4000 cells were captured for each condition, and cell counts were performed using the FIJI StarDist plugin [32]. Measurements for each pooled sample were conducted in duplicate, and pools showing greater than 60% neutralization were classified as positive for neutralizing activity.

### 2.5. Data Analysis

For the analysis of seroprevalence data, the mean signal for each sample was calculated after subtracting the background signal originating from the secondary antibody alone. To establish a threshold for positivity, we calculated the average background absorbance obtained from wells coated with non-infected cell lysates, adding three times the standard deviation of these values. ELISAs were performed under identical experimental conditions to those used for detecting adenovirus antibodies, but with non-infected cell lysates and using positive control sera. Each serum sample was then classified as either positive or negative for adenovirus antibodies based on the calculated threshold. Using this classification, seroprevalence percentages were computed to determine the distribution of positive samples across the studied population.

For neutralization data analysis, sera within pools classified as positive were considered to have effective neutralizing activity (see Appendix A).

Graphical representations were generated using R packages (version 4.5.0). Bar charts were created using the ggplot2 package (version 3.5.0) [33] while heatmaps were generated using the “pheatmap” package to illustrate the levels of seropositivity across the various adenoviruses analyzed [34].

Hierarchical clustering was applied to group viruses with similar antibody recognition profiles. Specifically, we used the complete-linkage method, which defines the distance between two clusters as the maximum distance between any pair of elements from the clusters. Distances between individual viruses were calculated using the Euclidean metric, which measures the straight-line distance between points in multidimensional space, considering the seropositivity levels across all samples. The resulting dendrograms illustrate how viruses cluster together based on overall similarity in immune recognition, allowing the identification of groups of adenoviruses with comparable seroreactivity patterns.

### 2.6. Phylogenetic Analysis

Sequences for adenoviral proteins II (hexon), III (penton) and IV (fiber) from different AdVs in fasta format were first aligned using the ClustalW versión 2.1 algorithm implemented in the R package “msa” version 1.41.0 [35]. Then the multiple sequence alignment was converted to an alignment object using the alignment function and the distance matrix was calculated using the dist.alignment function, both from the “seqinr” R package version 4.2-36 [36]. The phylogenetic trees were constructed utilizing the Bio-Neighbor Joining Method with the R package “ape” version 5.8-1 [37]. The resulting tree was plotted using the “ggtree” R package version 1.4.11 [38].

## 3. Results

### 3.1. Evaluation of Binding Antibodies Against HAdVs and SAdVs

To determine the seroprevalence of binding antibodies against HAdV species A–F (HAd-A12, HAdV-B14, HAdV-C5, HAdV-C6, HAdV-D26, HAdV-D36, HAdV-E4 and HAdV-F41) and SAdVs (SAdV21, SAdV25 and SAdV31), sera from 2488 healthy adult donors were analyzed. The reactivity varied significantly between the different HAdV types, as depicted in the heatmap showing seropositive and seronegative percentages (Figure 1A and Appendix A). The highest seropositivity rates were observed for HAdV-C5 and HAdV-C6 (98.5% and 98.7%, respectively). Unexpectedly, high percentages were also detected against HAdV-E4 (98.2%), SAdV-31 (97.8%), SadV-25 (94.4%), HAdV-B14 (93.3%), HAdV-F41 (90.4%), HAdV-D36 (78.2%) and HAdV-A12 (76.5%). In contrast low percentages of seropositivity were obtained for HAdV-D26 and SAdV-21 (14.5% and 10.4%, respectively).

The ELISA values obtained for each of the analyzed serum samples against the different AdV types (Figure 1C) were subjected to clustering analysis (Figure 1B). The analysis revealed three distinct clusters: species C (HAdV types 5 and 6) clustered together with SAdV-31 and HAdV-E4, while HAdV-D26 was grouped with SAdV-21, positioned near HAdV-A12 and HAdV-D36. Additionally, HAdV-B14 clustered closely with HAdV-F41 and SAdV-25.

### 3.2. Evaluation of Neutralizing Antibodies Against HAdVs and SAdVs

While informative, the total antibodies detected via ELISA differ from neutralizing antibodies (NAbs), which provide a better indication of specificity and protective immunity [39]. Therefore, neutralization assays were performed on pools of the 2488 serum samples, grouped according to absorbance values obtained from the ELISAs (Appendix A), as detailed in the materials and methods (Figure 2 and Appendix A).

As anticipated, lower percentages of positivity were observed when measuring NAbs compared to binding antibodies. Nonetheless, NAbs against species C adenoviruses (HAdV-C5, HAdV-C6) remained the most prevalent. The highest levels of NAbs were observed for HAdV-C5 (86%), HAdV-C6 (86%), HAdV-B14 (81.9%), and HAdV-F41 (78.4%), all of which clustered together (Figure 2B). For these viruses, the prevalence of NAbs was consistent with the results obtained with the ELISAs. A distinct cluster grouped HAdV-D36 (41.9%), HAdV-A12 (42.9%), and HAdV-E4 (41.4%). Interestingly, the percentage of positive samples for NAbs in the case of HAdV-D26 was higher than in the ELISAs (23.2% vs. 14.5%), which grouped with the SAdVs, SAdV-25, SAdV-31, and SAdV-21 (4.9%, 4.0% and 0%, respectively).

The high percentage of seropositive samples against the SAdVs types is likely to be due to cross-reactivity of binding antibodies produced against one or more of the HAdVs types. Phylogenetic analysis of capsid proteins showed relationships between SAdVs and human AdV types, supporting this observation (Figure 3). Sequences of proteins II (hexon), IV (fiber), and III (penton) from all 11 viruses were used for multiple sequence alignment. The genetic distances showed a close relationship between the amino acid sequences of SAdV-21 and HAdV-B14, SAdV-25 and HAdV-E4, and between SAdV-31 and species C HAdVs (C5 and C6).

Amino acid sequence analysis of the hexon, penton and fiber structural proteins was used to map previously reported B-cell epitopes onto multiple sequence alignments of all human adenovirus types used in this study [40,41,42,43,44,45,46] (Appendix A). For the hexon protein most of the epitopes described are located within the hypervariable regions. Interestingly, despite being located in variable domains, we observed a considerable level of conservation across many of the serotypes. This pattern may partially explain the cross-reactivity of antibodies between different adenovirus types, while the specific sequence differences are likely responsible for the type-specific immune responses that have been described. In the case of the penton protein the predicted epitopes also fall within variable regions; however, these do not appear to be equally conserved across the different types. Instead, they showed conserved sequence clusters that are more family-specific, which suggests that antibody cross-reactivity may be restricted to AdV species rather than across all types. For the fiber protein B-cell epitopes are less well characterized. The lack of clearly defined linear epitopes is likely due to the structural requirements of this protein, as recognition seems to depend on the protein’s trimerization resulting in conformational epitopes located at the surface of the knob domain. This structural constraint may explain why cross-reactive antibody responses against fiber are less predictable and may differ substantially from those observed for hexon or penton.

## 4. Discussion

### 4.1. Seroprevalence of AdV Antibodies Across Different Populations

A continuous and updated seroprevalence analysis of adenovirus immunity at a national scale and across specific susceptible cohorts is crucial for informing vaccine development, public health preparedness, and understanding epidemiological dynamics. This study provides an extensive assessment of seroprevalence and neutralizing antibodies (NAbs) against various human adenovirus (HAdV) and simian adenovirus (SAdV) types in healthy adults across all 32 states in Mexico. Our results reveal a high prevalence of binding antibodies, particularly against HAdV-C5 and HAdV-C6, which were detected in over 98% of the samples.

Geographic and socioeconomic factors further contribute to deviations from national averages in seroprevalence, with urban populations, high-density areas, or low socioeconomic communities potentially experiencing higher exposure rates compared to more rural regions. However, the similarity in seroprevalence patterns across geographically distinct regions of Mexico highlights the ubiquity of certain HAdV types, possibly linked to widespread natural exposure. The dynamic nature of adenovirus immunity and the potential for heterogeneous exposure highlights the need for targeted seroprevalence surveillance in specific cohorts, ensuring that adenoviral vector vaccines are tailored for each population’s unique immune landscape and effectively overcome barriers posed by pre-existing immunity.

These findings are consistent with previous data from other countries and regions, including the United States, South America, Asia, and Africa, where these species are highly prevalent [20,23,24,25,26,27,28]. HAdV-E4 binding antibody levels were detected at a 98.2% level, and those for human adenovirus 41 were similar in Asia (73.4% in healthy population) [47]. To date, there are only two reports of HAdV-E4 prevalence, one from Belgium and one from the USA, in healthy individuals, reporting prevalence rates of 17% and 46%, respectively. Our results show very high levels for HAdV-E4; however, with data from previous reports being so scattered, it is not easy to conclude the distribution in developing countries [21,25]. The prevalence for HAdV-D36 in developed countries is reported to be low, between 5% and 10%. In contrast, our data is comparable to levels from less developed countries such as those in Africa and Asia (48.7% and 34.7%, respectively).

A study by Zheng et al. investigated the seroprevalence of neutralizing antibodies against adenovirus types 14 and 55 in healthy adults in Southern China [48], with overall seropositivity rates of 24.8% for HAdV-B14 and 22.4% for HAdV-B55. Interestingly, the study found that NAb levels against HAdV-B14 and HAdV-B55 tended to increase with age, with lower prevalence among individuals under 20 years. Additionally, the blood type of donors influenced NAb prevalence, with individuals of blood type AB having higher seropositivity and NAb titers. These results suggest significant variability in pre-existing adenovirus immunity based on demographic factors such as age and blood type. Comparing these findings with our study in Mexico, we observe similarities in the age-related patterns of adenovirus seroprevalence, with high antibody prevalence noted among adult individuals. However, unlike the findings in Southern China, we did not include blood type and age as variables in our analysis, which might provide an important additional insight into adenovirus immunity in our population.

Our study focused on the adult population in Mexico, but understanding seroprevalence differences across age groups, such as children or immunocompromised individuals, is vital since exposure rates and immune responses vary significantly between cohorts. The seroprevalence studies in various countries highlight the importance of focusing on different age groups, particularly children, in the evaluation of adenovirus immunity. For instance, younger children may have low levels of adenovirus exposure, while older adults often show high seroprevalence due to cumulative lifelong exposure. Tian et al. reported low levels of herd immunity against multiple HAdV types in young children from Guangzhou, China, emphasizing the vulnerability of this population and the necessity for vaccine development targeted towards children [49]. Similarly, the study conducted in India by Appaiahgari et al. demonstrated that children below two years had significantly lower levels of HAdV-C5 neutralizing antibodies, which increases after the age of 18 months, suggesting that younger children could be suitable candidates for HAdV-based vaccines before they reach higher exposure levels [50]. Furthermore, the study from Nigeria by Idris et al. underscored the high seroprevalence of HAdV in children, along with significant molecular characterization, indicating the potential for targeting this age group for vaccination to control respiratory infections early on [51]. These findings emphasize a significant limitation of our study on adenovirus seroprevalence in Mexico, as it primarily focused on adults, and did not include younger age groups, particularly children who are a vulnerable population for adenovirus infections. Including children in future studies would provide a more comprehensive view of adenovirus immunity across all age groups and help in designing effective vaccine strategies suitable for the entire population.

The study by Ouoba et al. analyzed the prevalence of neutralizing antibodies (NAbs) against adenoviruses HAdV-C5, HAdV-D26, and HAdV-B35 in Burkina Faso and Chad, comparing healthy and HIV-infected individuals. Their findings showed a high seroprevalence for HAdV-C5, reaching up to 90% in HIV-infected groups. Interestingly, HAdV-D26 seroprevalence was also considerably high (~47%) in sub-Saharan populations, whereas HAdV-B35 had consistently low prevalence across all groups [52]. These results provide a significant point of comparison to our seroprevalence data in Mexico, highlighting regional differences in immunity against HAdV.

Mexico’s unique environment and population dynamics may give rise to regional HAdV variants that are not well characterized in the global literature. There is an emerging need to conduct comprehensive adenovirus surveillance and characterization studies in Mexico, which could uncover novel serotypes with unique immunological features. Identifying these novel variants could also help explain unexpected patterns in immune responses, such as unpredicted cross-reactivity or immunity gaps.

### 4.2. Considerations on Study Population and Generalizability

Our study utilized serum samples from healthy adult blood donors. As described in demographic studies from the United States, China, India and Mexico [53,54,55,56,57] blood donors often differ from the general population in age, sex, educational level, socioeconomic status and health behaviors. For example, young adult males and individuals with higher education are frequently overrepresented, while lower-income or comorbid individuals may be underrepresented.

Despite these differences, blood donor cohorts provide a reasonable proxy for assessing adenovirus exposure in healthy adults. This approach allows estimation of seroprevalence of different adenovirus genera while minimizing potential confounding due to underlying disease. However, the specific characteristics of blood donors may not fully capture the diversity of the general population. Consequently, our findings should be interpreted as representative of healthy adult populations motivated to donate blood, rather than all demographic groups. Future studies including more heterogeneous populations, as well as pediatric and immunocompromised cohorts, would allow a more comprehensive understanding of adenovirus immunity across the full spectrum of age and health status.

### 4.3. Implications of Seroprevalence Data for Vaccine Design

Adenovirus-based vaccines are widely used, yet pre-existing neutralizing antibodies against common adenoviral vectors, such as HAdV-C5, can significantly compromise vaccine efficacy, especially in populations with widespread prior exposure. As described in previous sections, when comparing the national prevalence percentages of both binding and neutralizing antibodies, there is a lower prevalence of neutralizing antibodies than binding antibodies for all viruses except HAdV-D26. HAdV-D26, a less prevalent HAdV type, showed only a 23.2% positivity rate for neutralizing antibodies, a value significantly higher than its ELISA positivity rate of 14.5% in our cohort. The reactivity levels of HAdV-D26 antibodies were similar to those reported in trials conducted in developed countries in North America and Europe [20,26]. The higher NAb levels compared to binding antibody levels may arise from differences in the sensitivity of the ELISA vs. neutralization assays and cannot be clearly interpreted without antibody titration in each case. This difference notwithstanding, HAdV-D26’s ability to induce neutralizing antibodies, even in populations with generally low seroprevalence, suggests it could continue to be a useful vector for HAdV-based vaccines. However, the seroprevalence after vaccination against the HAdV types that were widely used against COVID-19 should be assessed.

Furthermore, exploring the potential of other less common HAdV or completely novel vectors might provide alternatives that circumvent existing immunity. Studies comparing the immunogenicity of human and simian AdV in populations with high pre-existing immunity will also be essential for designing next-generation vaccines that can overcome the challenges presented by widespread adenovirus exposure. For example, a study by Zhao et al. analyzed the seroprevalence of neutralizing antibodies (NAbs) against human adenovirus type 5 (HAdV-C5) and chimpanzee adenovirus type 68 (CAdV-68) in cancer patients compared to healthy adults in China [58]. Their results demonstrated a significantly lower prevalence of NAbs against CAdV-68 (37.3%) compared to HAdV-C5 (68.5%) across all participants, suggesting that CAdV-68 may serve as a more suitable vector for vaccine development, particularly in populations with high pre-existing immunity to HAdV-C5.

In addition to strain-specific immune responses, our results suggest that antibodies elicited by one viral strain may exhibit cross-reactivity with other related strains. Based on the multiple sequence analysis and epitope sequence conservation (Appendix A), potential cross-reactivity could be anticipated, possibly even with adenovirus strains that we cannot currently predict or that are already circulating in the Mexican population. This cross-reactivity could influence both the interpretation of serological assays and the potential for broad protection, as conserved epitopes in structural proteins are often responsible for such cross-reactive antibody responses. Understanding these patterns is important for vaccine design and for evaluating population-level immunity, as individuals previously exposed to one strain might have partial immunity to related strains. Further investigation into the molecular basis of cross-reactivity could provide insight into the mechanisms of protective immunity and potential cross-protection among viral variants. Taken together, these findings support the idea that the degree of epitope conservation among adenoviruses is likely to underlie the observed antibody cross-reactivity, particularly for hexon.

Another important consideration is the cross-reactivity observed between HAdVs and SAdVs. The phylogenetic analysis supports the hypothesis that high levels of binding antibodies against SAdVs might reflect cross-reactive antibodies elicited by related HAdVs, such as HAdV-C5 or HAdV-C6, HAdV-B14, and HAdV-E4. The clustering of HAdV-D26 with SAdVs also implies potential cross-reactive immune responses between HAdVs and SAdVs.

Understanding this relationship might help predict the immune impact of using simian-derived adenoviral vectors, thereby aiding in the development of vaccines that can bypass HAdV immunity. Interestingly, in the case of HAdV-B14 and SAdV-21, their phylogenetic proximity does not seem to result in high levels of binding antibody cross-reactivity. These findings have critical implications for the use of SAdVs in vaccine design, as cross-reactivity may influence vaccine efficacy and safety profiles in diverse populations. For SAdVs, although there is insufficient data on the prevalence of these viruses in human populations, we suspect that the high levels detected in our assays could correspond to cross-reactivity with other AdVs, including those reported and not reported here. Given that AdVs have broad host ranges and tissue tropism, and cross-species infections of monkey and human viruses have been described [59,60].

The limitations of the study were as follows: (1) The analysis presented in this study was performed using pooled serum samples, which may obscure individual-level variations in immune response. This limitation could potentially lead to an underestimation or overestimation of antibody prevalence for certain AdV types. (2) A detailed analysis of the antibody titers of each individual serum, rather than pooled serum samples, would help to identify specific correlates of neutralization vs. binding antibodies. (3) We did not include blood type or age as variables in our analysis. (2) No samples were included in pediatric and adolescent populations. (3) Socio-economic conditions were unknown.

Additionally, detailed analysis of the immune response at the individual level would help to identify specific correlates of protection and variability in immune response to AdVs. This is particularly important for determining the suitability of different AdV vectors in diverse populations.

## Figures and Tables

**Figure 1 viruses-17-01184-f001:**
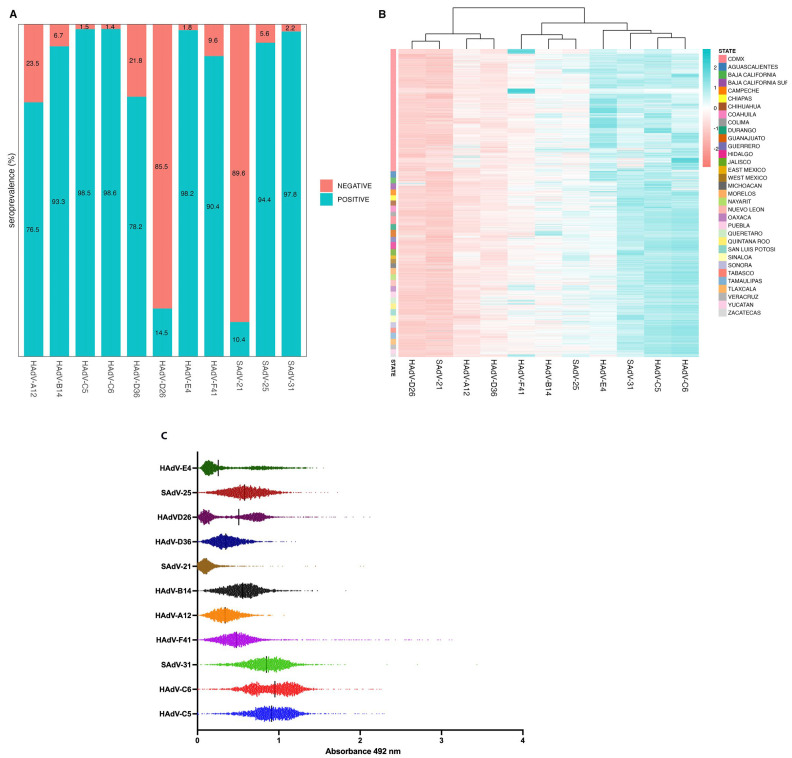
**Binding antibodies against human and simian adenoviruses in healthy Mexican adults.** ELISA immunoassays against the 11 indicated AdV types were performed for all 2488 serum samples of healthy adult volunteers. (**A**) Percentages of seropositive and seronegative samples for each adenovirus type. (**B**) Heatmap of z-normalized absorbance values for each sample across the tested adenoviruses, color-coded by intensity. The dendrogram above the heatmap illustrates clustering of viruses based on similarities in antibody responses. The color annotation on the right indicates the Mexican states from which the samples originate, allowing for the visualization of geographical patterns in seropositivity. (**C**) Binding antibody reactivity against 11 HAdV and SAdV types. Sera were diluted 1:100 in 1% milk in PBS-T and analyzed by ELISA as described in the Materials and Methods section. The plot shows the raw absorbance values at 492 nm for each of the 2488 serum samples. The seropositive threshold for each virus was calculated as the mean background signal plus three times the standard deviation (0.291), which was used as the cutoff for positivity. Values below this threshold were considered background.

**Figure 2 viruses-17-01184-f002:**
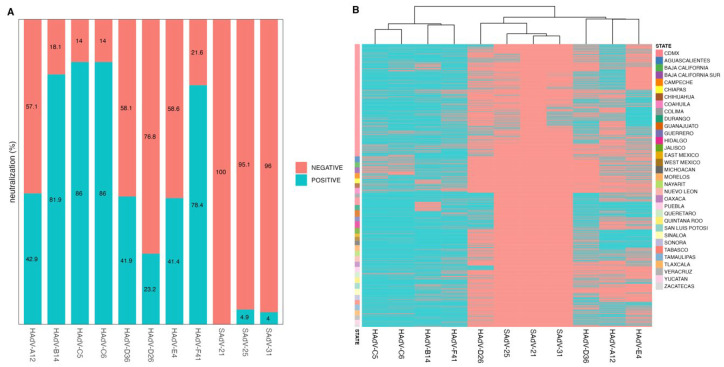
**Neutralizing antibodies against human and simian adenoviruses in healthy Mexican adults.** The prevalence of neutralizing antibodies was assessed using neutralization assays performed on pools of sera, grouped according to the absorbance values obtained from the ELISA for each of the 11 indicated HAdV types (see materials and methods). (**A**) Percentage of sample pools with neutralizing antibodies for each virus. (**B**) Heatmap showing the presence of neutralizing antibodies across the samples and viruses, with positive pools colored in green and negative ones in orange, based on a 60% neutralization threshold (see Appendix A). The dendrogram above shows clustering of viruses by similarity in neutralization patterns. The row color annotation on the left represents the Mexican states from which the samples originated, illustrating regional differences in neutralizing antibody prevalence.

**Figure 3 viruses-17-01184-f003:**
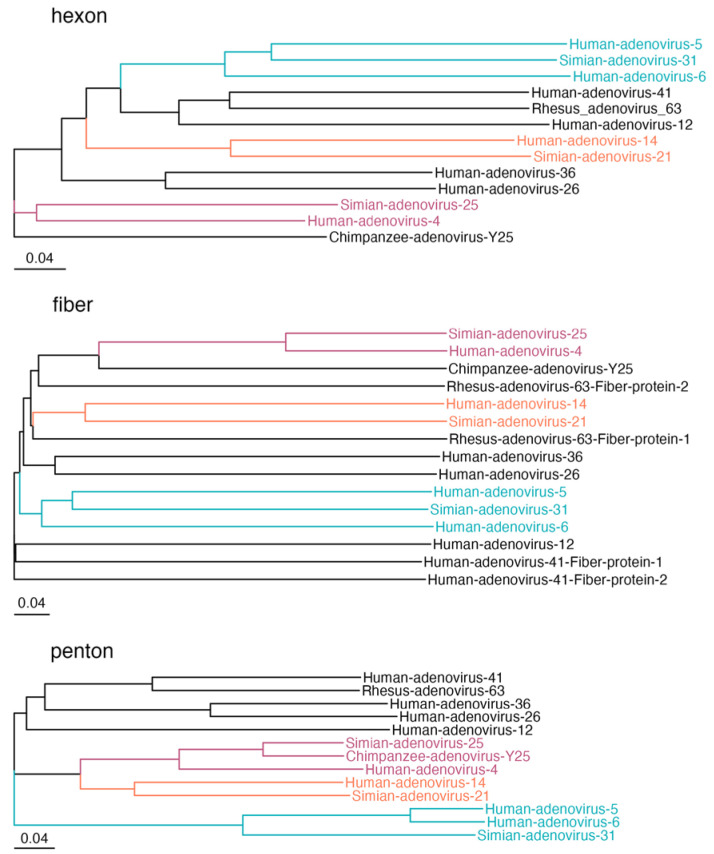
**High levels of binding antibodies against SAdVs could arise from cross-reactivity with HAdV antibodies.** Phylogenetic trees for adenovirus capsid proteins hexon, fiber and penton. Trees were constructed using the Bio-neighbor joining method (see materials and methods). Color-coded branches show AdVs that are closely related to SAdV21 (orange), SAdV25 (magenta) and SAdV31 (cyan). The scale bars represent the same genetic distance for each of the phylogenetic trees.

## Data Availability

The original contributions presented in this study are included in the article/Appendix A. Further inquiries can be directed to the corresponding author.

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
