# Peer review of "Seroprevalence of Neutralizing Antibodies in Healthy Adults, in Mexico, Against Human and Simian Adenovirus Types"

_viruses, 2025, doi:10.3390/v17091184_

Round 1
Reviewer 1 Report
Comments and Suggestions for Authors
This is a very interesting paper on seroprevalence of adenoviruses in Mexico. The study design appears to be sound, and the paper is well-written, with appropriate conclusions.
Unfortunately, the supplemental figures/tables were not available for me to review.
I have a few minor suggestions for additional discussion points. The methods state that donors met the criteria for blood donation - what are these criteria? Were any additional data on these blood donors available for analysis? I understand that the dataset is quite limited, but analysis of age group and sex would be interesting if available. Is there an upper limit for age for blood donors? Are there any papers that exist that describe demographics of blood donors? That might be an interesting point to make - in addition to being healthy adults, I would think that blood donors likely differ from the larger population in other ways, e.g., socioeconomic status.
Also, the serum samples were collected in two different time periods - was there a difference in seroprevalence over time?
Author Response
Response to Reviewer 1 Comments
We sincerely thank the reviewer for their evaluation of our manuscript, and for their comments and suggestions. Please find the responses to comments below. The corresponding revisions have been included in the modified manuscript highlighted in blue.
This is a very interesting paper on seroprevalence of adenoviruses in Mexico. The study design appears to be sound, and the paper is well-written, with appropriate conclusions.
Comments 1: Unfortunately, the supplemental figures/tables were not available for me to review.
Response 1: We have made sure the supplementary figures and tables were uploaded separately and with the manuscript pdf so they are readily available.
I have a few minor suggestions for additional discussion points.
Comments 2: The methods state that donors met the criteria for blood donation - what are these criteria?
Response 2: The criteria for blood donation were those established by the Mexican Official Norm (NOM-253-SSA1-2012) and the official procedures for attention to blood donors at the Blood Banks of the Mexican Institute of Social Security (Number. 2430-003-001, dated November 24th, 2020), and have been included in the modified manuscript.
Comments 3: Were any additional data on these blood donors available for analysis?
Response 3: For the samples obtained from the CNTS, data are available for blood type, hemoglobin and hematocrit levels; however, no information on donor age or sex is available for these samples. For the samples obtained from the IMSS, data are only available for age and sex. This limited and inconsistent availability of demographic data across the two sources does not allow for a detailed analysis of these factors and their potential influence on adenovirus seroprevalence, as such an analysis would introduce bias.
Comments 4: I understand that the dataset is quite limited, but analysis of age group and sex would be interesting if available. Is there an upper limit for age for blood donors?
Response 4: As mentioned above the donors included in this study met the standard criteria for blood donation at each institution. For the CNTS samples, no data on age or sex is available, so we cannot specify an upper age limit, other than the age established by the criteria for blood donation. For the IMSS samples, age and sex information is available, and upper age limit was 65. Given that complete data are not available for all samples, it is not possible to perform a detailed analysis by age group or sex in this study. This is certainly an interesting point and should be addressed in future studies with controlled donor groups. A short section addressing this limitation of our study has been added to the modified manuscript.
Comments 5: Are there any papers that exist that describe demographics of blood donors? That might be an interesting point to make - in addition to being healthy adults, I would think that blood donors likely differ from the larger population in other ways, e.g., socioeconomic status.
Response 5: In our study, serum samples were obtained from blood donors, who represent ostensibly healthy adults. As documented in several demographic studies of blood donors (Patel et al., 2019; studies in China, India, and Mexico), donors often differ from the general population in aspects such as age, sex, educational level, socioeconomic status, and health behaviors. For example, young adult males with higher education are frequently overrepresented among donors, whereas individuals with lower income or comorbidities may be underrepresented.
Despite these demographic differences, we consider the samples appropriate for estimating the prevalence of different adenovirus genera in healthy individuals, as donors reflect a generally healthy adult population. This approach allows for inference of exposure patterns and adenovirus circulation while minimizing potential bias from concurrent illnesses that could influence antibody presence.
At the same time, we acknowledge the inherent limitation of our study population: the specific characteristics of blood donors may not fully represent the diversity of the general population across all demographic or socioeconomic strata. Therefore, our results should be interpreted in the context of a healthy, motivated adult donor population and not necessarily generalized to all population segments. Future studies in more heterogeneous cohorts could complement these findings to more broadly assess adenovirus circulation across different population strata. The short section added to the modified manuscript addresses this important point.
Comments 6: Also, the serum samples were collected in two different time periods - was there a difference in seroprevalence over time?
Response 6: Although the serum samples were collected in two different time periods, our analysis showed only minor differences in the seroprevalence. This allowed us to combine the two groups and present them as a single population. Doing so increased the statistical power of our analysis. Furthermore, since all CNTS samples were from Mexico City, this approach enabled us to perform a geographical analysis by different city zones, as shown in the supplementary material.
Reviewer 2 Report
Comments and Suggestions for Authors
Studying the immune response to adenoviruses is important not only for understanding the prevalence of different variants but also for vaccine development. Adenoviruses are promising candidates for developing adenovirus vaccines. However, a pre-existing immune response may negatively impact vaccine efficacy. Therefore, strains that are rare in the population should be used for vaccine development. The prevalence of total and neutralizing antibodies against different types of adenoviruses in Mexico is estimated here.
Some comments are listed below:
1. Line 204. Why were these virus strains chosen? Please specify.
2. Line 217. Please describe in more detail how the clusters were identified.
3. Figure 1C is unclear. Please explain what this figure shows. What value was considered seropositive. What value is background?
4. Paragraph 3.1. Here, neutralizing activity was estimated for all samples. It is obvious that seronegative samples lack virus-specific IgG. Such antibodies bind the virus and can neutralize it. Therefore, neutralizing activity should be analyzed separately for seropositive samples.
5. Lines 260-266. What is the percentage of homology between the capsid proteins of the viruses? Please provide data on the similarity between the viruses and the possibility of antibody cross-reactivity. Which epitopes of the viral proteins have regions of homology?
6. Discussion section. Please add more information on the cross-reactivity of antibodies to different virus strains.
Author Response
Response to Reviewer 2 Comments
We sincerely thank the reviewer for their evaluation of our manuscript, and for their comments and suggestions. Please find the responses to comments below. The corresponding revisions have been included in the modified manuscript highlighted in blue.
Studying the immune response to adenoviruses is important not only for understanding the prevalence of different variants but also for vaccine development. Adenoviruses are promising candidates for developing adenovirus vaccines. However, a pre-existing immune response may negatively impact vaccine efficacy. Therefore, strains that are rare in the population should be used for vaccine development. The prevalence of total and neutralizing antibodies against different types of adenoviruses in Mexico is estimated here.
Some comments are listed below:
Comments: 1. Line 204. Why were these virus strains chosen? Please specify.
Response 1: The virus types were selected in order to include at least one common type for which previous reports exist and information on their prevalence is available from studies worldwide from each HAdV species, and that were immediately available from the ATCC at the time the study was initiated. Consequently, various relevant HAdV, such as the species G type 52, species B1 types 3 and 7, and species B2 type 11 could not be included in the study. In addition, the three simian adenovirus types were selected because they are closely related to groups B, C and E, because of their similarity to AdV used in vaccines, and their availability from ATCC. A short description for the choice of virus species and types has been included in the materials and methods section of the modified manuscript.
Comments: 2. Line 217. Please describe in more detail how the clusters were identified.
Response 2: For the analysis of seroprevalence data, the mean signal for each sample was calculated after subtracting the background signal originating from the secondary antibody alone. To establish a threshold for positivity, we calculated the average background absorbance obtained from wells coated with non-infected cell lysates, adding three times the standard deviation of these values. ELISAs were performed under identical experimental conditions to those used for detecting adenovirus antibodies, but with non-infected cell lysates and using positive control sera. Each serum sample was then classified as either positive or negative for adenovirus antibodies based on the calculated threshold. Using this classification, seroprevalence percentages were computed to determine the distribution of positive samples across the studied population.
For neutralization data analysis, sera within pools classified as positive were considered to have effective neutralizing activity (see Supplementary Figure 1).
Graphical representations were generated using several R packages. “ggplot2” was used to create bar charts [1] while heatmaps were generated using the "pheatmap" package to illustrate the levels of seropositivity across the various adenoviruses analyzed [2].
Hierarchical clustering was applied to group viruses with similar antibody recognition profiles. Specifically, we used the complete-linkage method, which defines the distance between two clusters as the maximum distance between any pair of elements from the clusters. Distances between individual viruses were calculated using the Euclidean metric, which measures the straight-line distance between points in multidimensional space, considering the seropositivity levels across all samples. The resulting dendrograms illustrate how viruses cluster together based on overall similarity in immune recognition, allowing the identification of groups of adenoviruses with comparable seroreactivity patterns.
A short description of this point has been included in the in the materials and methods section of the modified manuscript.
Comments: 3. Figure 1C is unclear. Please explain what this figure shows. What value was considered seropositive. What value is background?
Response 3: Figure 1C shows the raw absorbance values at 492 nm for each of the 2,488 serum samples analyzed. The seropositive threshold for each virus was calculated as the mean background signal plus three times the standard deviation, resulting in a threshold value of 0.291. This value represents the cutoff for positivity for each virus, while signals below this threshold were considered background. These details have been added to the figure legend for clarity.
Comments: 4. Paragraph 3.1. Here, neutralizing activity was estimated for all samples. It is obvious that seronegative samples lack virus-specific IgG. Such antibodies bind the virus and can neutralize it. Therefore, neutralizing activity should be analyzed separately for seropositive samples.
Response 4: Neutralizing activity was analyzed independently by pooling samples according to their absorbance levels at 492 nm. This approach allowed us to group the highest (positive) values separately from the lowest (negative) values. In this way, we were able to differentiate the neutralization activity of seropositive sera and identify which samples lacked neutralizing activity. Additionally, to avoid nonspecific neutralization caused by unrelated antibodies, several dilutions were tested using both known negative and positive samples. From these experiments, we determined that a 1:200 dilution allowed for accurate measurement of neutralizing activity without producing false-positive neutralization in seronegative samples.
Comments: 5. Lines 260-266. What is the percentage of homology between the capsid proteins of the viruses? Please provide data on the similarity between the viruses and the possibility of antibody cross-reactivity. Which epitopes of the viral proteins have regions of homology?
Response 5: Data has now been included in the modified manuscript (Table S3) with the percentage of amino acid sequence identity between 13 AdV types for the hexon, penton and fiber proteins. Also, we have included multiple sequence alignments of these three proteins, displaying the B-cell epitopes that have been previously reported (Fig. S2), and sequence identity matrices for each of the epitope sequences (Table S5). A short section describing the potential contribution of sequence similarity to antibody cross-reactivity has been added to the Results section.
Comments: 6. Discussion section. Please add more information on the cross-reactivity of antibodies to different virus strains.
Response 6: In addition to type-specific immune responses, our results suggest that antibodies elicited by one viral strain may exhibit cross-reactivity with other related strains. Based on the alignment of reported immunogenic peptides, we can anticipate potential cross-reactivity and possibly even with adenovirus strains that cannot currently be predicted to be circulating in the Mexican population. This cross-reactivity could influence both the interpretation of serological assays and the potential for broad protection. Previous studies have shown that conserved epitopes in structural proteins are often responsible for such cross-reactive antibody responses. Understanding these patterns is important for vaccine design and for evaluating population-level immunity, as individuals previously exposed to one strain might have partial immunity to related strains. Further investigation into the molecular basis of cross-reactivity could provide insight into the mechanisms of protective immunity and potential cross-protection among viral variants. This point has been added to the discussion in the modified manuscript.
Round 2
Reviewer 2 Report
Comments and Suggestions for Authors
The new additions to the manuscript made a big difference. The quality of the paper had improved, and all my questions were addressed. No more comments.